# Left-Behind Children’s Subtypes of Antisocial Behavior: A Qualitative Study in China

**DOI:** 10.3390/bs12100349

**Published:** 2022-09-21

**Authors:** Wen Liu, Weiwei Wang, Lingxiang Xia, Shuang Lin, Yining Wang

**Affiliations:** 1Department of psychology, Liaoning Normal University, No.850, Huanghe Road, Dalian 116029, China; 2Department of psychology, Xinan University, No.2 Tiansheng Road, Chongqing 400715, China

**Keywords:** antisocial behavior, left-behind children, qualitative, subtypes

## Abstract

As a result of the recent decades of urbanization and industrialization, millions of people have migrated to cities in search of better work opportunities in China. Meanwhile, their children have often been left behind in the care of other family members. To classify the subtypes of antisocial behavior of the left-behind children, this qualitative study interviewed a total of 71 participants, including five groups: left-behind children, parents, teachers, principals and community workers. The findings showed that left-behind children’s antisocial behavior is manifested as the type of limited adolescent antisocial behavior, and the three subtypes of left-behind children’s antisocial behavior were rule-breaking behavior, delinquent behavior and criminal behavior. In addition, the development of children’s antisocial behavior could range from general violations to delinquent behaviors and even to criminal behaviors.

## 1. Introduction

Over the recent decades of urbanization and industrialization, millions of people migrated to cities in search of better work opportunities in China. Migrant workers generally cannot move their families to cities because of the rural–urban separation, poor work conditions and their own financial constraints [1,2]. As a result, children are often left behind in the care of other family members [3]. In this study, left-behind children (LBC) refers to children aged 3 to 16 whose parents work outside the hometown or one of them works outside, and the other is unable to fulfill the parenting responsibilities. They stay at their original residence for at least 6 months. [4].

According to a survey of National women’s federation of children’s-work department, in 2014, 22% of children (about 61 million) were left at home by their migrant parents. Generally, the children are left in the care of extended family members. A recent survey study suggested that 74% of LBC lived with their grandparents, 12.8% lived with their uncles/aunts, and 13.2% were left to take care of themselves [5].

Family has an irreplaceable effect on the development of LBC. If there are many disadvantages in these domains (e.g., insecure parental attachment, poor parental supervision and low parental support), LBC may, as a result of a lack of effective social support, be more likely to engage in various socially disapproved behaviors in attempts to attract the others’ attention. Parental absence as a result of long-term rural-to-urban migration causes great disruption to the family structure, the family’s emotional functions and parental supervision of children’s behaviors [6,7,8]. Being separated from their parents, LBC often lack family care and effective parenting. Meanwhile, they are faced with risks, such as insufficient family care, inadequate family education, communicating with undesirable peer groups and being susceptible to malicious information on the internet. There is accumulating evidence that the LBC have been found, on average, to show higher risk of depression [9,10], smoking [11], being more prone to developing problem behaviors (such as fighting, smoking and alcohol abuse) and delinquent behavior [12,13]. Obviously, the disadvantaged background of the family has a negative impact on the development of LBC.

Antisocial behavior refers to behaviors that violate societal norms and the personal or property rights of others [14], and there are multiple types of antisocial behavior. For instance, Burt and his colleagues suggested that the subtypes of antisocial behavior included physically aggressive, rule-breaking and socially aggressive behavior [15]. Meanwhile, previous studies have measured children’s antisocial behavior in terms of types. Mishra assessed the trends in antisocial behavior over the past year by three subtypes of antisocial behavior (mild, moderate and severe) [16]; Mann classified juvenile antisocial behavior into minor and serious infractions according to its consequences [17]; Cook divided juvenile antisocial behavior into four categories: non-antisocial behavior, assault, theft and serious antisocial behavior [18].

However, no research had been published on the subtypes of LBC’s antisocial behavior (a preliminary literature search was carried out in April 2020 through Web of Science using the search terms: ‘left-behind children’, ‘antisocial behavior’ and ‘subtypes’, searching for articles). That is, presently, there are few studies on the antisocial behavior of LBC compared with the general developing children. Meanwhile, the evaluation of LBC’s antisocial behavior should be based on the current historical background they live in. Therefore, this study aimed to find the salient themes of LBC’s subtypes of antisocial behavior with a qualitative approach.

## 2. Method

### 2.1. Participants

In order to make the sample as homogenous as possible, we targeted one rural area in the northeast of China and one rural area in the southwest of China, which are two typical rural areas with big populations of LBC with similar economic developmental level. In each area, participants were recruited from one primary school, one secondary school and two communities where the schools were located. All participants (*n* = 71) signed a consent form, and the researcher gathered their background information.

There were five participant groups: left-behind children, parents, teachers, principals and community workers. The LBC group included 16 primary and secondary school students aged 10~16. Both of their parents had left the hometown to work in other cities over 12 months and only went back home less than 3 times per year. The parents group included the 12 fathers/mothers of LBC, and their age ranged from 37 to 44. The teachers group included 33 teachers/directors of LBC, and their age range was 30–51. The principals group included 3 principals aged 44~49 of the schools we selected, and their working responsibilities were caring for and helping LBC. The community workers group included 7 chairmen of women’s federations aged 35–64 who had engaged in children and women’s work more than ten years. More information is shown in Table 1.

### 2.2. Interviews

Before the formal interviews, pre-interviews with 6 individuals from every group were conducted, and we adapted the language style and the way of raising the questions, particularly for the LBC themselves. In the formal interviews, a 50~60 min interview via telephone, video call or in person was conducted individually. The individual interview method was used to carry out the research. This method guaranteed greater individual privacy and made more real the thoughts, deep attitude and personal experience expressed and could explore the nature of the questions [19]. A semi-structured interview protocol was undertaken broadly based on the key research questions. The questions were open ended, and the interview schedule had 7 questions (Table 2).

Through the first question, the interviewer could clarify the main goal of the study and help participants understand the meaning of antisocial behavior in psychology. The second question gave the interviewer the opportunity to help participants construct the structure of children’s antisocial behavior and facilitate answering the following questions. Through the questions 3 to 5, we obtained the information about LBC’s specific antisocial behavior.

### 2.3. Data Collection and Analysis

The interviews were conducted from May to November 2020 during the epidemic. The interviews with children, principals and some of the teachers were conducted in a mental health classroom or a meeting room of their school. The interviews with other teachers, parents and community workers were conducted by phone or online meeting call, from May to July. Although it is suggested that in-person interviews might be more advantageous, nevertheless, it is suggested that there is no significant difference between phone interview and face-to-face interview. [20]. Given the consent from every participant, the entire process of every interview was recorded by two devices in case of device failure. The participants received CNY 80 for their time and effort. All the recordings were transcribed and rechecked verbatim by the authors directly after the interview.

The recordings were transcribed within 48 h of the interview to ensure accurate transcription. All the transcripts were managed and further coded by Nvivo12.0 (QSE International Pty Ltd., Burlington, MA, USA). The coding of the interviews was conducted according to the grounded theory [21]. The grounded theory, as an evolving qualitative research method, can integrate the strengths inherent in quantitative methods with qualitative approaches [22]. Its goal is generating theory together with its completeness of method. Therefore, it can be distinguished from other qualitative methods.

There are three steps of coding: open coding, focused coding and theoretical coding. In open coding, under the principle of maintaining openness, accuracy, conciseness and other grounded theories, 10 case texts were named in terms of words, sentences and events to form the original codes. In focused coding, we identified the most important and frequently original codes. These codes could explain more texts and formed the generic codes. In theoretical coding, the relationships between the generic codes were specified and formed the integrated theoretical codes. The authors were divided into two groups and conducted the coding simultaneously.

### 2.4. Results

When quoting the interview data, the following conventions were used:

[…] Where text has been removed.

(…) A pause.

This study aimed to understand the subtypes of antisocial behavior of LBC. Therefore, the findings from this study are presented in three parts according to the three subtypes. Specifically, LBC’s antisocial behavior can be divided into three subtypes, namely, rule-breaking behavior, delinquent behavior and criminal behavior. The development of children’s antisocial behavior can range from general violations, delinquent behavior and even to criminal behavior.

In all the interviews, behavior associated with breaking rules/delinquent behavior/criminal behavior of LBC was mentioned 92/27/6 times. The results showed that LBC’s antisocial behavior was common, relatively transient, situational and near normative, but not rare, persistent or pervasive. It should be noted that adolescence is the peak period of antisocial behavior in LBC (see Figure 1).

### 2.5. Subtype 1: Rule-Breaking Behavior

Participants distinctly described that the primary antisocial behavior of LBC was rule-breaking behavior (*n* = 42)—for example, lying, aggressive behavior, not finishing homework on time, sleeping in class, cheating in an exam, disrupting the class and truancy playing hooky.

Teacher Meng: Some of LBC copy other people’s homework when they arrive at school in the morning. That is, they don’t do their homework when they go home […] I think it’s because their guardians are unable to fulfill all of the parenting responsibilities. I’ve heard from another teacher that one LBC in her class (aged 13, girl) often make noises and delay to hand in homework in the class. 

Teacher Meng described some typical rule-breaking behaviors of LBC in their class. The above performance is reflected in rule-breaking behavior. For more information, we analyzed more information from interviews with LBC, their parents and other teachers.

Teacher Ann: LBC communicate less with other students at school. When a conflict happened, some of them will show a higher level of aggressive behavior. Also, I’ve heard something about school bullying.

Left-behind child Zhao: I borrow my classmate’s homework to copy. If they won’t lend it to me, I hit them.

Interviewer: What will your parents do if they know about your copying?

Left-behind child Zhao: They won’t find out. They only come home once a year.

Parent May: His (LBC, his son and aged 14) academic performance is bad. He thinks it’s hard to study, and he doesn’t want to study anymore […].

Interviewer: What did you say when he said ‘I don’t want to study anymore?’

Parent May: ‘You have to study hard, so that you can find a good job in the future.’

We classified rule-breaking behavior as type I antisocial behavior because it has a higher frequency among LBC’s antisocial behavior. In all the interviews, LBC’s behavior associated with breaking the rules was mentioned 92 times. Specifically, participants described that some LBC’S rule-breaking behaviors included aggression behavior, school bullying, cheating, violation of discipline and so on.

### 2.6. Subtype 2: Delinquent Behavior

Participants described that the secondary antisocial behavior of LBC was delinquent behavior (*n* = 15).

Teacher Zhou: One of the LBC (14-year-old boy) in my class robbed others of their money.

Interviewer: Do you know why he took the money?

Teacher Zhou: He thinks other people have more money than he does. 

Interviewer: Has he ever had any rule-breaking or problem behaviors before?

Teacher Zhou: Yes. He has had fights with others and skipped school last year.

Left-behind child Rong: I don’t usually go home after school.

Interviewer: Why don’t you go home after school?

Left-behind child Rong: There’s nobody at home.

Interviewer: What do you do after school?

Left-behind child Rong: Once, I asked a younger student for money, and I beat him because he didn’t give me the money. A few days ago, a LBC stole a shared bike after school.

Interviewer: Do you know that stealing a shared bike is illegal behavior?

Left-behind child Ray: I have no idea.

(Note: Shared bikes can provide the right to use a bike for all people. People can unlock the shared bikes by simply using their smartphone. Problems such as illegal parking, vandalism and shared bike theft are illegal behaviors in China. In 2018, a man who stole a shared bike was sentenced to a 3-month detention with a 3-month probation and fined CNY 1000 by the Shanghai Minhang People’s Court.)

Compared with type I antisocial behavior (rule-breaking behavior), type II antisocial behavior (delinquent behavior) occurred less frequently in some LBC. However, type II antisocial behavior is illegal or delinquent. Behavior associated with delinquent behavior of LBC was mentioned 27 times. Participants suggested that LBC’s delinquent behavior included vandalism, fight with other people, steal money from other people, pilfer public property and so on. Moreover, some LBC with delinquent behavior always showed some rule-breaking or problem behavior before.

### 2.7. Subtype 3: Criminal Behavior

Participants described that the third grade of antisocial behavior of LBC was criminal behavior (*n* = 3). Subtype III of LBC’s antisocial behavior is criminal behavior, which includes robbery, intentional injury, murder and a series of behaviors that violate the national law.

Teacher Chen: I have heard about some LCB’s criminal behavior from other teachers. One of his LBC students (Z, aged 16, boy) took a dagger in his bag when he went to school. He and his LBC group often grab money after school. Once, he had an argument with others and stabbed a boy with his dagger.

Interviewer: What does ‘LBC group’ mean?

Teacher Chen: A gang of children who have left behind experience. They robbed other classmates’ cash.

Type III antisocial behavior is the highest level of LBC’s antisocial behavior. Behaviors of type III antisocial behavior may carry serious outcome. Unfortunately, this type of behavior is more likely to occur in the left-behind groups. A Report of the Supreme People’s Court of China suggested that LBC accounted for 70% of the crimes committed by all minors in 2013.

In our interviews, LBC behavior associated with criminality was mentioned six times. Based on the data, some LBC are mainly involved in violent crimes, such as robbery and intentional injury, which account for about 45% and 36% of the total number of crimes. From the data, it could be concluded that the violent tendency of LBC’s crimes is very obvious. Among the crimes committed by LBC in our interview, 70% were organized crimes, and the age of criminal gangs was younger. From the interviews, it was suggested that crimes committed by some LBC were often committed together, with a certain degree of coordination, and some of them formed fixed criminal groups. 

## 3. Discussion

The goal of the current study was to provide evidence for the subtypes of LBC’s antisocial behavior via a qualitative approach. Meanwhile, our findings provide theoretical contributions to the development of LBC and the design of a self-report assessment. To accomplish this task, we collected the data from interviews and reports of specific instantiations over nearly two months via in-person, telephone and online interviews. A total of 71 participants included five groups: LBC, parents of LBC, teachers who have LBC in their classes, principals who have great populations in their schools and community workers with a wealth of experience in dealing with LBC issues. They were interviewed regarding LBC’s antisocial behavior, which included the aspects of internal states and social outcomes. Our results suggest that LBC’s antisocial behavior is manifested as limited adolescent antisocial behavior. Three subtypes of LBC’s antisocial behavior includes rule-breaking behavior, delinquent behavior and criminal behavior. In addition, the development of children’s antisocial behavior could range from general violations to delinquent behaviors and even to criminal behaviors.

### 3.1. Antisocial Behavior of LBC Is Adolescence-Limited

The antisocial behavior of LBC was manifested as limited adolescent antisocial behavior in the samples of this study. Moffitt and his colleagues outlined two hypothetical prototypes: life-course persistent versus adolescence-limited antisocial behavior [23]. To be specific, life-course persistent individuals’ antisocial behavior has its origins in neurodevelopmental processes, beginning in the childhood, building persistently thereafter and continuing into midlife. In contrast, adolescence-limited delinquents’ antisocial activities have their origins in age-graded social processes that begin with a maturity gap in adolescence and end when social adulthood is attained [24]. In the samples of this study, the antisocial behavior of LBC was manifested as adolescence-limited antisocial behavior. In all the interviews, LBC’s antisocial behavior was common, relatively transient, situational and near normative, but not rare, persistent or pervasive. It was indicated that the antisocial behavior of LBC was adolescence limited.

Like previous studies, temporary, situational antisocial behavior is quite common among the population, especially among adolescents. Moreover, these different behaviors appear to have similar developmental trajectories. Rule-breaking behavior is most prevalent in primary school time, and older LBC showed more rule-breaking behavior. Delinquent and criminal behaviors are quite infrequent during childhood and increase dramatically over the course of adolescence. All three behaviors fall off by early adulthood.

### 3.2. Three Subtypes of LBC’s Antisocial Behavior

Antisocial behavior of LBC could be classified into three subtypes. The first subtype of LBC’s antisocial behavior is rule-breaking behavior, which includes lying, attacking others, not finishing homework on time, sleeping in class, cheating in an exam, disrupting the class, truancy and so on. The second subtype of LBC’s antisocial behavior is delinquent behavior, which includes stealing, vandalism, substance abuse, fighting and brawls. The third subtype of LBC’s antisocial behavior is criminal behavior, which includes robbery, intentional injury, murder and a series of behaviors that violate the national law. 

The subtypes of antisocial behavior are very meaningful because the framework of generality for deviance suggests that various forms of antisocial behavior tend to co-occur among individuals [25]. Specifically, rule-breaking behavior is the most prevalent antisocial behavior, which occurs with the highest frequency among LBC. Delinquent and criminal behaviors, by contrast, are quite infrequent. Additionally, it was suggested that the earlier the manifestation of rule-breaking behavior of LBC appears, the more profound the development of antisocial behavior will be, even developed to criminal behavior, which will inevitably have a negative impact on the development of LBC, their families, schools and the society. Therefore, prevention of and attention to antisocial behavior of LBC should start as early as possible at a younger age, and attention should be paid to the behavioral manifestations of children, especially the first subtype of antisocial behavior. Nonetheless, more work is needed to firmly ground these typology findings. More typology work in the future should be conducted to cement these three subtypes of antisocial behavior within the literature.

### 3.3. Special Characteristics of LBC’s Antisocial Behavior

Compared with children in general, the antisocial behavior of LBC showed some special characteristics. First, the rule-breaking behavior of LBC was often associated with a lack of parental discipline. For example, some LBC interviewees told us they did not finish homework on time or skipped school because ‘parents = migrated to other cities to work’. Secondly, the delinquent and criminal behaviors of LBC were mainly robbery and injury cases. Third, LBC’s delinquent and criminal behaviors were often group organized, and the age characteristic tended to be younger.

The ecological systems theory could explain the special characteristics of LBC’s antisocial behavior. It is suggested that individuals’ development is influenced by their interaction with micro systems (i.e., family atmosphere, school environment). Therefore, LBC’s antisocial behaviors are mostly the product of the comprehensive effects of family, school and other social environments. Specifically, long-term parental migration may lead to changes in the family structure and seriously diminish basic family affection and connections. For example, the parent–child cohesion reduces the risk of rural left-behind children adopting problematic behaviors, such as smoking and drinking [6,13]. Furthermore, cumulative ecological risks have a detrimental effect on children’s behaviors [26]. As we know, LBC may face risks such as lack of family supervision, educational resources and the problem of peer interaction. More precisely, LBC’s antisocial behavior outcome is not caused by a single risk factor but by a synergy of multiple interrelated risk factors [27]. That is, in terms of these cumulative risks, left-behind children are at a higher risk than non-left-behind children [28,29].

Previous studies also suggested that cumulative ecological risk can significantly predict tobacco/alcohol use and aggressive behaviors among adolescents and young children [30,31]. Thus, cumulative ecological risk constitutes a new perspective for advancing our understanding of antisocial behavior among Chinese LBC. In China, several researchers have noted that the problem behaviors of LBC are shaped by the overlapping of multiple risk factors rather than by a single risk factor [32]. However, not all LBC developed antisocial behaviors. There are individual differences regarding antisocial behavior. Some individuals from the left behind group adjust well to their situation and show resilience and better self-esteem characteristics, despite exposure to multiple risk contexts [6,13]. In contrast, LBC with a high-frequency rule-breaking behavior always show callous–unemotional traits, deficits in empathy, guilt and pro-sociality [33]. Moreover, researchers highlight the association between a positive family environment and the development of the trait of compassionate social emotions above [34,35].

### 3.4. Suggestions for Intervention of LBC with Antisocial Behavior

Daily practice and training of metacognition for LBC with antisocial behavior were suggested. This could identify the non-operating established habits and replace them with more functional and useful ones to achieve what is called self-accomplishment through brain rewiring and brain development [36]. Specifically, Drigas and Mitsea present the eight pillars of metacognition, including academic and theoretical knowledge of cognition and cognitive abilities, operational knowledge about the functionality of cognitive abilities, self-monitoring, self-regulation, adaptation, recognition–anagnorisis, discrimination–diakrisis and mnemosyne. Among eight of them, self-regulation is more related to the individual’s antisocial behavior. Because successful self-regulation depends mainly on top-down control and suppressing the occurrence of antisocial behavior.

Meanwhile, the prevention of antisocial behavior also plays an important role in LBC’s development. Antisocial behavior is more likely to be caused when metacognitive skills are not properly trained. LBC’s physical, intellectual, emotional and spiritual intelligence could be improved by improving the metacognitive skills. That is, successful metacognitive skills guarantee LBC’s personal, academic and professional success, emotional well-being and social adjustment.

### 3.5. Limitations

There are several limitations to the current study. The first is that we relied on a convenience sample of primary and middle school students and their parents/teachers. However, it is important to note that the sample in this study still represents a sort of convenience sample, with the resultant limitations. The sampling approach in this study stands in contrast to superior methods involving the recruitment of participants through random sampling methods. Further studies should examine whether truly random samples exhibit results consistent with this study. Second, the research examining the nature and extent of participation in antisocial behavior (ASB) in typically developing individuals during late adolescence and early adulthood remains rare. A need for ongoing longitudinal research in typically developing samples was highlighted, particularly on the transition to adulthood [37].

## 4. Conclusions

This qualitative study interviewed a total of 71 participants, including five groups: left-behind children, parents, teachers, principals and community workers. Qualitative analysis of the transcripts indicated that:LBC’s antisocial behavior is manifested as adolescence-limited antisocial behavior, especially among adolescents;LBC’s antisocial behavior is classified into three subtypes, which includes rule-breaking behavior, delinquent behavior and criminal behavior;The development of LBC’s antisocial behavior can range from general violations to delinquent behavior and even to criminal behavior.

## Figures and Tables

**Figure 1 behavsci-12-00349-f001:**
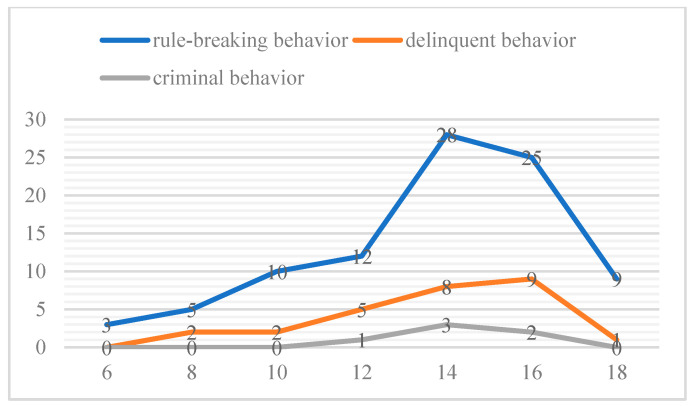
Frequency of antisocial behavior in children of different ages.

**Table 1 behavsci-12-00349-t001:** Numbers of participant in each group.

	Primary School	Secondary School	Total
Children	7	9	16
Parents	6	6	12
Teachers	12	21	33
Principals	2	1	3
Community workers	—	—	7
Total			71

**Table 2 behavsci-12-00349-t002:** Interview schedule.

Interview Questions
**The Basic Information**
1. What is your overall understanding of children’s antisocial behavior?
2. What are the specific manifestations of children’s antisocial behavior?
3. What are the characteristics, strategies and performance of left-behind children who have antisocial behavior?
4. Could you please talk about a specific example of left-behind children’s antisocial behavior?
5. What are the influencing factors of left-behind children’s antisocial behavior?
6. What has been, if anything, challenging or negative about teaching/parenting left-behind children for you so far?
7. Is there anything else that you would like to comment on with regard to left-behind children’s antisocial behavior?

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
