# Peer review of "Left-Behind Children’s Subtypes of Antisocial Behavior: A Qualitative Study in China"

_behavsci, 2022, doi:10.3390/bs12100349_

Round 1

Reviewer 1 Report

This qualitative study interviewed 71 participants to classify the sub-types of left-behind children’s antisocial behavior. The methods and the results are very meaningful and make contributions to left-behind children’s behaviour researches. Minor revisions are needed of this paper:

(1) Please revise the in-text citation and references format to in line with the style of this journal, which is ACS format.

(2) Please let a native speaker to help with the editing.

(3) Please summarize the theoretical contributions with one or two sentences in the discussion.

Author Response

Response to Reviewer 1 Comments

Comments and Suggestions for Authors

This qualitative study interviewed 71 participants to classify the sub-types of left-behind children’s antisocial behavior. The methods and the results are very meaningful and make contributions to left-behind children’s behaviour researches. Minor revisions are needed of this paper:

We are grateful to reviewer for his/her effort reviewing our paper and his/her positive feedback. The summary of our work as written by this reviewer is precise. Here below we address the questions and suggestions raised by the reviewer #1.

(1) Please revise the in-text citation and references format to in line with the style of this journal, which is ACS format.

Response:This suggestion is appreciated. The in-text citation and references format problems in the original manuscript was improved in the revised manuscript.

(2) Please let a native speaker to help with the editing.

Response:We apologize for the language problems in the original manuscript. The language presentation was improved with assistance from a native English speaker with appropriate research background.

(3) Please summarize the theoretical contributions with one or two sentences in the discussion.

Response:We are grateful for the suggestion. To be more clearly and in accordance with the reviewer concerns, we have added a more detailed interpretation regarding the theoretical contributions. 

Page 6-part of discussion-paragraph 1-line 2~3.

“The goal of the current study was to provide evidence for the subtypes of LBC’s antisocial behavior via a qualitative approach and to make theoretical contributions to the development of LBC.”

Reviewer 2 Report

The topic of the article is important and interesting. The research undertaken is original. I positively assess the method used.

Remarks:

1. The method, technique and tool should be described in more detail.

2. It should be presented in more detail with the grounded theory.

3. It should be presented in more detail with the grounded theory as presented by Kathy Charmaz. It is very important.

4. I suggest detailing the discussion.

I consider the development of a typology to be valuable.

Author Response

Response to Reviewer 2 Comments

The topic of the article is important and interesting. The research undertaken is original. I positively assess the method used.

We have carefully addressed reviewer's concerns. Please see below our replies. We hope he/she is satisfied with our answers we provided. Changes highlighted in red have been made accordingly in the revised manuscript.

Remarks:

  1. The method, technique and tool should be described in more detail.

Response:This suggestion is appreciated. It has been clarified in the revised version of the manuscript. Page 3- paragraph 3-line 6~9 and page 4- paragraph 1-line 1~4

“Although it is suggested that in-person interviews might be more advantageous, however, it is not identified a significant difference between the results of the interviews conducted by phone and in-person (Sturges & Hanrahan, 2004).”

“The recordings were transcribed within 48 hours of the interview to ensure accurate transcription. All the transcripts were managed and further coded by Nvivo12.0 (QSE International Pty Ltd, Burlington, MA, US) to facilitate the coding and ensure the consistency of coding process”

  1. It should be presented in more detail with the grounded theory. It should be presented in more detail with the grounded theory as presented by Kathy Charmaz. It is very important.

Response:We are grateful for the suggestion. To be more clearly and in accordance with the reviewer concerns, we have added a more detailed interpretation regarding the theoretical contributions. More detailed statistical analysis was added on page 4- paragraph 1-line 6~12.

“The coding of interviews was conducted according to the Grounded Theory (Charmaz, 2006). Grounded theory, as an evolving qualitative research method, could integrate the strengths inherent in quantitative methods with qualitative approaches (Walker & Myrick, 2006). Specifically, researchers using grounded theory set out to gather data and then systematically develop the theory derived directly from the data (Dey, 1999). Moreover, grounded theory could be distinguished from other qualitative methods because of its goal of generating theory together with its completeness of method.”

  1. I suggest detailing the discussion.

I consider the development of a typology to be valuable

Response:We are grateful for the suggestion. To be more clearly and in accordance with the reviewer concerns, we have added a part of conclusion in page 8- paragraph 1-line 3~5

“Nonetheless, more work is needed to firmly ground these findings of typology. More work of typology in the future should be done to cement these three subtypes of antisocial behavior within the literature.”

Reviewer 3 Report

This qualitative study aims to classify and provide evidence regarding the subtypes of left-behind children’s antisocial behavior. Interviews from a total of 71 participants including five groups: left-behind children, parents, teachers, principals, and community workers were analyzed. The findings demonstrated that left-behind children's antisocial tendency appears as adolescent limited antisocial behavior. In addition, rule-breaking conduct, delinquent behavior, and criminal behavior were the three categories of left-behind children's antisocial behavior.

Indeed, the researchers address an important and challenging topic for modern societies which has been overlooked. The purpose is clearly stated. The applied methodology is adequately described. The language is clear. The style is academic, but at the same time reader oriented. The abstract gives the reader a good idea of what to expect from the paper. The headings enable the reader to understand the main points of the paper and follow its structure. The tables illustrate important points.

I would like to suggest some minor revisions. Although conclusions are not mandatory, it would be useful to include a small paragraph summarising the main conclusions with a critical eye, because the discussion is quite long. In addition, in the discussion, it would be interesting to make a brief mention regarding the importance of self-regulation as an important asset in developing inner strengths and self-regulated behaviors. Metacognition can play an important role in the prevention as well as in the intervention of antisocial behaviors (see Drigas and Mitsea, 2020; 2021). According to the guidelines, references must be numbered in order of appearance in the text.

Drigas, A., & Mitsea, E. (2020). The 8 pillars of metacognition. International Journal of Emerging Technologies in Learning (iJET), 15(21), 162-178.

Author Response

Response to Reviewer 3 Comments

We appreciate reviewer #3 for his/her effort to review our manuscript, and his/her positive feedback. The reviewer gives an accurate summary of our work and brings forward constructive questions. We have addressed them below.

(1) I would like to suggest some minor revisions. Although conclusions are not mandatory, it would be useful to include a small paragraph summarising the main conclusions 

Response:We are grateful for the suggestion. To be more clearly and in accordance with the reviewer concerns, we have added a part of conclusion in page 9.

“This qualitative study interviewed a total of 71 participants including five groups: left-behind children, parents, teachers, principals, and community workers. Qualitative analysis of transcripts indicated that: LBC’s antisocial behavior is manifested as adolescence limited antisocial behavior, especially among adolescents; LBC’s antisocial behavior is classify into three subtypes, which including rule-breaking behavior, delinquent behavior and criminal behavior; The development of LBC's antisocial behavior can range from general violations, to delinquent behaviors, and even to criminal behaviors.”

(2) with a critical eye, because the discussion is quite long. In addition, in the discussion, it would be interesting to make a brief mention regarding the importance of self-regulation as an important asset in developing inner strengths and self-regulated behaviors. Metacognition can play an important role in the prevention as well as in the intervention of antisocial behaviors (see Drigas and Mitsea, 2020; 2021). According to the guidelines, references must be numbered in order of appearance in the text.

Drigas, A., & Mitsea, E. (2020). The 8 pillars of metacognition. International Journal of Emerging Technologies in Learning (iJET), 15(21), 162-178.

Response:We are grateful for the suggestion. As suggested by the reviewer, we have added more details of the intervention and prevention for LBC, which including metacognition ability and self-regulation.

Page 9- part of Suggestions for intervention of LBC with antisocial behavior.

3.3. Suggestions for intervention of LBC with antisocial behavior

“Daily practice and training of metacognition for LBC with antisocial behavior were suggested. It could identify the non-operating established habits and replace them with more functional and useful ones, to achieve what is called self-accomplishment through brain rewiring and brain development (Drigas &Mitsea, 2020). Specifically, Drigas and Mitsea (2020) present the eight pillars of meta-cognition, including academic and theoretical knowledge of cognition and cognitive abilities, operational knowledge about the functionality of cognitive abilities, self-monitoring, self-regulation, adaptation, recognition-anagnorisis, discrimination-diakrisis and mnemosyne. Among 8 of them, self-regulation is more related to indivisual’s antisocial behavior. Because self-regulation constitutes a set of ongoing, dynamic, and adaptive modulations by oneself. A successful self-regulation depend mainly on top-down control and suppress the occurrence of antisocial behavior. Meanwhile, the prevention of antisocial behavior also play a important role in for LBC’s development. It is more likely to cause antisocial behavior when metacognitive skills are not properly trained. LBC’s physical, intellectual, emotional and spiritual intelligence could be improved by improving metacognitive skills. That is, a successful metacognitive skills guarantee LBC’s personal, academic and professional success, emotional well-being and social adjustment.”